# A Partial Ambiguity Resolution Algorithm Based on New-Breadth-First Lattice Search in High-Dimension Situations

**DOI:** 10.3390/s22197126

**Published:** 2022-09-20

**Authors:** Shouhua Wang, Zhiqi You, Xiyan Sun, Libo Yuan

**Affiliations:** 1School of Information and Communication, Guilin University of Electronic Technology, Guilin 541004, China; 2Guangxi Key Laboratory of Precision Navigation Technology and Application, Guilin University of Electronic Technology, Guilin 541004, China; 3National Engineering Research Center for Satellite Navigation, Positioning and Position Service, Guilin 541004, China; 4School of Optoeletronic Engineering, Guilin University of Electronic Technology, Guilin 541004, China

**Keywords:** RTK sensor networks, partial ambiguity resolution, lattice theory, high-dimension situations

## Abstract

With the development and integration of GNSS systems in the world, the positioning accuracy and reliability of GNSS navigation services are increasing in various fields. Because the current multisystem fusion leads to an increase in the ambiguity dimension and the ambiguity parameters have discrete characteristics, the current conventional search algorithm leads to low search efficiency when the ambiguity dimension is large. Therefore, this paper describes a new algorithm that searches the optimal lattice points by lattice theory through the breadth-first algorithm and reduces the search space of ambiguity by calculating and judging the Euclidean distance between each search variable and the target one so as to propose a new lattice ambiguity search algorithm based on the breadth-first algorithm. The experimental results show that this method can effectively improve the search efficiency of ambiguity in high-dimension situations.

## 1. Introduction

With the rapid development of GNSS navigation and positioning services, the positioning accuracy of the GNSS navigation and positioning system is required more and more in both military and civil fields. Therefore, determining how to realize the high-precision positioning of the GNSS navigation and positioning system with high reliability has become a core problem. Now, the multisystem fusion leads to an increase in the ambiguity dimension, and the ambiguity parameters have discrete characteristics. Therefore, to search the integer ambiguity solution, it is necessary to build the corresponding search space and then use the corresponding search algorithm to search and fix them in the search space so as to obtain the integer ambiguity solution parameters. However, the conventional search algorithms lead to low search efficiency when the ambiguity dimension is large.

Due to the strong correlation between the ambiguity parameters in high dimensions, it is necessary to reduce the correlation of the ambiguity parameters and then use the appropriate search algorithm to search and fix them in the search space. Teunissen [1] proposed the least-squares ambiguity reduction adjustment (LAMBDA) search algorithm. It reduces the correlation of ambiguity parameters and then searches and fixes them in the search area, similar to an ellipsoid, so as to obtain the fixed value of the integer solution of ambiguity. Chang [2] reduced the correlation of the lambda algorithm according to lattice theory, combined it with the greedy selection algorithm and the inverted transformation algorithm to eliminate the ambiguity parameters, and proposed the MLAMBDA algorithm, which reduced the search space and improved the fixation efficiency.

At present, there are mainly two kinds of methods to solve the ambiguity based on lattice theory. One method, on the basis of lattice basis specification, considereds that the perfect orthogonal relationship between basis vectors can be realized, and the solution of the problem can be obtained by direct rounding; the other method is to obtain the solution of the problem by constructing a certain search space. For the solutions to these two kinds of problems, the former method is represented by the nearest plane method, which is simple in principle and easy to implement, but it is an approximate method, and the results depend on the effect of lattice protocol. The later kind of method is represented by the spherical search algorithm, which is called the maximum-likelihood (ML) decoding algorithm in the field of communication [3,4]. Teunissen also calls this kind of method the integer least-squares algorithm. Compared with the approximate method, it is more complex, but it is the best solution method for this discrete problem at present.

## 2. RTK Sensor Networks

### 2.1. Sensor Model

Generally, the pseudo range equation and carrier phase observation equation can be established for a receiver, r, and a satellite, i:(1)$$\rho_{r}^{i}=R_{r}^{i}+c\delta t_{r}-c\delta t^{i}+I_{r}^{i}+T_{r}^{i}+\xi_{\rho ,r}^{i}$$
(2)$$\varphi_{r}^{i}=\lambda^{-1}\left(R_{r}^{i}+c\left(\delta t_{r}-\delta t^{i}\right)-I_{r}^{i}+T_{r}^{i}\right)+N_{r}^{i}+\xi_{\varphi ,r}^{i}$$

Among them, $\rho_{r}^{i}$ and $\varphi_{r}^{i}$ represent the pseudo-range and carrier phase observation, respectively, $I_{r}^{i}$ represents the ionospheric delay, $\xi_{\rho ,r}^{i}$ represents the pseudo-range observation noise, $\xi_{\varphi ,r}^{i}$ represents the carrier phase observation noise, $\delta t_{r}$ represents the receiver clock difference, $T_{r}^{i}$ represents the pair flow delay, c represents light speed, $R_{r}^{i}$ represents the satellite ground distance between the receiver and the satellite, $\delta t^{i}$ represents the satellite clock difference, and $N_{r}^{i}$ represents the integer ambiguity of the carrier phase, which can be represented by receiver coordinate $\left(x,y,z\right)$ and satellite coordinate $\left(x^{i},y^{i},z^{i}\right)$:(3)$$R_{r}^{i}=\sqrt{{\left(x-x^{i}\right)}^{2}+{\left(y-y^{i}\right)}^{2}+{\left(z-z^{i}\right)}^{2}}$$

In GNSS differential positioning, the single differential positioning model refers to the difference between different receivers for the same satellite. Assuming that ground reference stations are r and b and observation satellites are j and s at the same time, differential processing is carried out between reference stations r and b. Relative to satellite s, the observation equation is:(4)$$\Delta \rho^{s}=\Delta R^{s}+c\Delta \delta t+\Delta I^{s}+\Delta T^{s}+\Delta \xi_{\rho }^{s}$$
(5)$$\Delta \varphi^{s}=\lambda^{-1}\left(\Delta R^{s}+c\Delta \delta t-\Delta I^{s}+\Delta T^{s}\right)+\Delta N^{s}+\Delta \xi_{\varphi }^{s}$$

Based on the single-difference model, the difference to eliminate the error between receivers is called the double-difference model, that is, the intersatellite difference between satellite s and satellite j. As shown in Figure 1.

Therefore, after two differences, the double-difference observation equations are:(6)$$\nabla \Delta \rho =\nabla \Delta R+\nabla \Delta I+\nabla \Delta T+\nabla \Delta \xi_{\rho }$$
(7)$$\nabla \Delta \varphi =\lambda^{-1}\left(\nabla \Delta R-\nabla \Delta I+\Delta T\right)+\nabla \Delta N+\nabla \Delta \xi_{\varphi }$$
where, “∇” refers to the intersatellite difference operator.

After the secondary difference equation between the monitoring station and the satellite, the satellite clock difference and the receiver difference are completely eliminated in the observation equation, and the ionospheric delay error and tropospheric delay error are weakened due to their strong correlation. Generally, when the baseline distance between the two receivers is not too long, the atmospheric error has a certain correlation, which can be eliminated by the double-difference model, so the observation equation can be simplified:(8)$$\nabla \Delta \rho =\nabla \Delta R+\nabla \Delta \xi_{\rho }$$
(9)$$\nabla \Delta \varphi =\lambda^{-1}\nabla \Delta R+\nabla \Delta N+\nabla \Delta \xi_{\varphi }$$

### 2.2. Ambiguity Resolution

At present, the LAMBDA is mostly used to fix ambiguity. This algorithm is a method to fix the ambiguity search domain [5,6,7].

Therefore, the integer ambiguity solution formula based on the LAMBDA algorithm is as follows:(10)y=Aa+Bb+ξ
where, y is the observation after the double-difference model, A is the ambiguity coefficient matrix, B is the coefficient matrix of non-ambiguity matrix parameters, a is the integer ambiguity parameter, b is the non-ambiguity parameter vectors, and ξ is the noise variable of the observation data.

The search objective function is the distance between the floating-point ambiguity solution and the integer ambiguity solution. The search is carried out through the predetermined ambiguity search space. When the objective function takes the minimum value, it means that the ambiguity integer solution is solved successfully, that is:(11)$$\underset{a}{\min}{\left(\tilde{a}-a\right)}^{T}Q_{\tilde{a}\tilde{a}}^{-1}\left(\tilde{a}-a\right)$$

The LAMBDA method specifies the following search space when searching [8,9,10,11]:(12)$${\left(\tilde{a}-a\right)}^{T}Q_{\tilde{a}\tilde{a}}^{-1}\left(\tilde{a}-a\right)<T$$
where T is the threshold of the search space. All integer values within the search range are the objects to be searched.

After the double-difference model, there is a certain correlation between the ambiguity parameters. The existence of a correlation will lead to a long, time-consuming, and low efficiency search process of the integer solution. Therefore, before using the lambda algorithm to search and fix the integer ambiguity solution, it is necessary to reduce the correlation [2,12,13]. This operation is usually called the Z transformation, and its process is as follows:(13)$$\tilde{z}=Z^{T}\tilde{a}$$
(14)$$Q_{\tilde{z}\tilde{z}}=Z^{T}Q_{\tilde{a}\tilde{a}}Z$$

Among them, the reduced correlation matrix, Z, should meet the following three conditions: (1) matrix Z should be a full-rank matrix with positive and inverse transformation characteristics; (2) $\left|\det Z\right|=1$; and (3) matrix Z should have the property of an integer so as to ensure that it is also an integer value after the table conversion.

At this time, Equation (6) correspondingly becomes:(15)$${\left(\tilde{z}-z\right)}^{T}Q_{\tilde{z}\tilde{z}}^{-1}\left(\tilde{z}-z\right)<T$$

After the search results are obtained, the final integer ambiguity solution can be obtained through the Z transformation.

## 3. Ambiguity Search Algorithm Based on Lattice Theory

### 3.1. Transition Relationship

There is a famous question about the research of lattices [14]: for a known point $y\in R^{n}$ in a lattice, it may not be on a lattice point, but the point is located in the lattice space, and there must be a lattice point x with the shortest distance from point y in a space. The expression is:(16){||y−x||≤||y−α||,∀α∈Λ}
where grid point x is the nearest grid point to y. That is, the nearest grid point search problem is the problem of solving the point x. Figure 2 is a schematic diagram in two-dimensional linear space.

According to Equation (14), since the estimated covariance matrix, $Q_{\tilde{a}\tilde{a}}^{-1}$, is a positive definite matrix, according to Cholesky it can be decomposed into:(17)$$Q_{\tilde{a}\tilde{a}}^{-1}=R^{T}R$$

At the same time:(18)$$\underset{a}{\min}{\left(\tilde{a}-a\right)}^{T}Q_{\tilde{a}\tilde{a}}^{-1}\left(\tilde{a}-a\right)$$

This can be converted to:(19)$$\underset{a}{\min}{\left(\tilde{a}-a\right)}^{T}Q_{aa}^{-1}\left(a-\tilde{a}\right)a\tilde{a}={\left(Ra-R\tilde{a}\right)}^{T}(R\tilde{a}-Ra)$$

This equation assumes that $R\tilde{a}=y$ is ultimately available:(20)$$\underset{a}{\min}{\left(\tilde{a}-a\right)}^{T}Q_{aa}^{-1}\left(a-\tilde{a}\right)a\tilde{a}=\min||Ra-y|{|}^{2}$$

At the same time, according to the nearest grid point search problem of Equation (10), for a given point of $y\in R^{n}$ in linear space, the grid point x closest to point y is searched in the grid diagram. Suppose that there is a number $\alpha \in Z^{k}$. Then, for x∈Λ,x=Rα, it can be described as bellow:(21)$$||y-x|{|}^{2}=||y-R\alpha |{|}^{2}=\underset{\forall \alpha \in \wedge }{\min}||y-c|{|}^{2}=\underset{\forall \alpha \in \wedge }{\min}||y-R\alpha |{|}^{2}$$

From Equations (10) and (14), it can be seen that the search for the integer ambiguity solution based on the least-square method is equivalent to the search for the grid point with the shortest distance from a known point in the grid diagram. Therefore, the search for the optimal ambiguity solution can be transformed into the search for the grid point with the shortest distance.

### 3.2. Ambiguity Estimation

In the conventional LAMBDA search algorithm, it is necessary to reduce the correlation of the ambiguity parameters. When it is converted to the search problem of a point in the lattice, the corresponding correlation reduction algorithm also needs to be adopted. Therefore, the ambiguity covariance matrix is decomposed by a triangular decomposition method and is then orthogonally transformed by the Gram-Schmidt process to realize the function of correlation reduction. First, the covariance matrix of the ambiguity parameter can be decomposed into:(22)$$\begin{array}{l} Q_{aa}^{-1}=R^{T}R\\ Q_{aa}^{-1}=V^{T}F^{T}FV \end{array}$$
where R is the full-rank matrix. From Equation (16), if matrix R can be transformed into an orthogonal matrix, then $Q_{aa}^{-1}$ is a diagonal matrix:(23)$$h_{i}^{*}=h_{i}-\sum_{j=1}^{i-1}\mu_{ij}h_{j}^{*}$$
where $h_{i}$ is the column vector of matrix R, $h_{i}^{*}$ is the orthogonalized vector, $\mu_{ij}=\frac{<h_{i},h_{j}^{*}>}{<h_{j}^{*},h_{j}^{*}>}$ is the conversion coefficient, and <⋅,⋅> represents the inner product value of two vectors. Since the ambiguity is an integer, the coefficients need to be rounded during orthogonalization, so the transformation process of matrix vector can be written as follows:(24)$$f_{i}=h_{i}-{\sum_{j=1}^{i-1}\left[\frac{<h_{i},f_{j}>}{<f_{j},f_{j}>}\right]}_{round}f_{j}$$
where $f_{j}$ is the transformed vector. Due to rounding, $f_{i}$, $f_{k}$, and k=1⋅⋅⋅i−1 may not be completely orthogonal. Therefore, the included angle between $f_{i}$ and the previously calculated vector can be infinitely close to the right angle through the iterative method, that is:(25)$$f_{i}^{k}=f_{i}^{k-1}-{\sum_{j=1}^{i-1}\left[\frac{<f_{i}^{k-1},f_{j}>}{<f_{j},f_{j}>}\right]}_{round}f_{j}$$
where $f_{i}^{k}$ is the value of vector $f_{i}$ transformed after k rounds of orthogonalization and $f_{i}^{0}=h_{i}$. When $|\frac{<f_{i}^{k-1},f_{j}>}{<f_{j},f_{j}>}|\leq 0.5$, stop the iteration and make $f_{i}=f_{i}^{k}$. The above orthogonalization process can be expressed in a matrix as bellow:(26)R=FV
where $F=[f_{1},f_{2},\cdot \cdot \cdot ,f_{n-1},f_{n}]$ and $V=V_{n}V_{n-1}\cdot \cdot \cdot V_{2}V_{1}$ are integer transformation matrices, while $V_{i}=V_{i}^{k}V_{i}^{k-1}\cdot \cdot \cdot V_{i}^{1}$.
(27)$$V_{i}^{k}= [\begin{array}{cccccccccc} 1 & & & & & v_{\left(1,i\right)}^{k} & & & & \\ & 1 & & & & \cdot & & & & \\ & & \cdot & & & \cdot & & & & \\ & & & \cdot & & \cdot & & & & \\ & & & & \cdot & v_{\left(i-1,i\right)}^{k} & & & & \\ & & & & & 1_{\left(i,i\right)} & & & & \\ & & & & & & \cdot & & & \\ & & & & & & & \cdot & & \\ & & & & & & & & \cdot & \\ & & & & & & & & & 1 \end{array}]$$

In the above formula, $v_{\left(i-1,i\right)}^{k}={\left[\frac{<f_{i}^{k-1},f_{i-1}>}{<f_{i-1},f_{i-1}>}\right]}_{round}$ is the coefficient corresponding to iteration $f_{i-1}$ of the k iteration when the i column vector is orthogonalized as an integer, and $V_{i}$ is obtained after the superposition of k iterations. Therefore, the variance matrix can be obtained, which is expressed by F and V:(28)$$\begin{array}{l} Q_{aa}^{-1}=V^{T}F^{T}FV\\ Q_{aa}^{-1}=UDU^{T} \end{array}$$

Since F is obtained by an integer orthogonal transformation, $F^{T}F$ is closer to the diagonal matrix than $Q_{aa}^{-1}$.

In recent years, lattice theory has been widely studied and applied in the fields of convex body analysis, number theory, computer science, communication [15], and cryptography. Many excellent processing methods have been developed for the two problems on lattices. Considering the equivalence between the fixed ambiguity problem and the above intermediate Sauger search, the relevant theory of lattices can be introduced into the ambiguity solution. By constructing the lattice space associated with the ambiguity search and then converting the search for the integer ambiguity solution into the search for the nearest point on the lattice, the specific process can be designed as follows:
(1)Due to the increase in the satellite altitude angle, the tropospheric delay decreases gradually. Therefore, combined with the partial ambiguity fixing strategy, the satellite cut-off altitude angle is set, and the satellites with altitude angles less than 15° are eliminated to obtain the optimal subset;(2)Ignoring the constraint of the ambiguity integer, the real ambiguity solution, $\overset{\wedge }{a}$, and its corresponding covariance matrix, $Q_{aa}^{-1}$, can be obtained by processing the optimal subset through an extended Kalman filter;(3)The lattice Λ(B) is related to ambiguity and is constructed based on the covariance matrix. Then, the lattice base, B, can be obtained from $Q_{aa}^{-1}=B^{T}B$ through the triangular decomposition of the covariance matrix, and the target vector, $y=B\overset{\wedge }{a}$, is established. From then on, the ambiguity search can be transformed into the search of lattice Λ(B) and known target vector y;(4)According to the definition of lattice theory, the main goal of the fixed ambiguity solution is to find the vector $l=B\alpha ,\alpha \in Z^{n}$ on the lattice that can meet the requirements of Equation (12). Since the lattice base, B, is known, the solution of the lattice is actually to solve the group of integer coefficient vectors that generate l. For the lattice Λ(B) and target vector y constructed in the second step, this vector is the unknown ambiguity vector.

After these four steps, we can obtain the solution of integer ambiguity. The third step is the key link in the lattice-based ambiguity solution. The later content of this paper focuses on this problem.

### 3.3. Breadth-First-Based Ambiguity Search Algorithm

Viterbo introduced the spherical search algorithm in the application of signal decoding [2,16,17], but because the spherical search algorithm needs to set the initial radius in advance, the size of the radius has a great impact on the efficiency and the success rate of the solution. If the radius is too large, the correct solution can be guaranteed, but it increases the burden of calculation; if the radius is too small to improve the search efficiency, it may lead to the incorrect solution. To solve this problem, the breadth-first search method, commonly used in communication coding, is applied to the spherical search algorithm [18,19]. The search algorithm flow chart is shown in Figure 3.

The basic idea of the breadth-first search mode is to obtain a set of candidate sequences through an i=n,n-1,⋅⋅⋅,1 cyclic search and then update the search space and continue the new search until the optimal solution of the problem is obtained. The algorithm sets a search radius, ρ, in advance. Then, i=n,n-1,⋅⋅⋅,1 searches layer by layer. When searching for each layer, it first selects the lower bound value in the candidate value sequence, then orders i=i−1 to bring ${\overset{\wedge }{a}}_{i},\cdot \cdot \cdot ,{\overset{\wedge }{a}}_{n}$ into the next layer to continue the operation. If there is no candidate value in the next layer, it returns to this layer, reselects ${\overset{\wedge }{a}}_{i}={\overset{\wedge }{a}}_{i}+1$, then calculates the next layer. After several cycles, when i=1, a set of solutions, $\overset{\wedge }{a}=[{\overset{\wedge }{a}}_{1},\cdot \cdot \cdot {\overset{\wedge }{a}}_{n}]$, can be obtained, calculating the distance between the corresponding lattice vector, $B\overset{\wedge }{a}$, and the target vector and comparing it with the search radius. If less, the search radius can be updated to:(29)$$\rho =||y-B\overset{\wedge }{a}||$$

At this point, it makes i=n and starts a new round of search based on this radius. This cycle is repeated until the search radius can no longer be reduced.

The iterative process of the conventional breadth-first search algorithm is complex, and the efficiency of this algorithm gradually decreases with an increase in dimension. Therefore, based on the algorithm, the search process is optimized and innovated.

In the process of the i=n,n-1,⋅⋅⋅,1 layer by layer circular search, it calculates the partial Euclidean distance (PED) between each vector and the floating-point solution vector:(30)$$T_{i}(\overset{\wedge }{a})=\sum_{j=i}^{n}\overset{-}{(y}-\sum_{k=j}^{n}r_{i,j}{\overset{\wedge }{a}}_{j}{)}^{2}$$

It sorts the calculated Euclidean distance values. For example, when searching the i layer, a t≤K candidate ambiguity sequence $a_{1},\cdot \cdot \cdot a_{k}$ is calculated, including $a^{j}={\left[0,\cdot \cdot \cdot ,a_{i+1}^{j},\cdot \cdot \cdot ,a_{n}^{j}\right]}^{T},j=1,\cdot \cdot \cdot ,t$. Since $T_{i}(\overset{\wedge }{a_{x}})$ is the smallest, the subsequent ambiguity search process is based on $T_{i}(\overset{\wedge }{a_{x}})$, where t represents the Euclidean distance. At this time, it is necessary to calculate the corresponding value of each ambiguity candidate value, $a_{i}$:(31)$$e_{i}={\left(\overset{-}{y}-r_{i,k}a_{i}-\sum_{k=i+1}^{n}r_{i,k}a_{i}\right)}^{2}$$

The size if the ambiguity candidate value is calculated and judged based on whether it meets the following requirement:(32)$$T_{i+1}(\overset{\wedge }{a})+e_{i}\leq \rho^{2}$$

If it meets the requirements, the sequence $a^{j}={\left[0,\cdot \cdot \cdot ,a_{i+1}^{j},\cdot \cdot \cdot ,a_{n}^{j}\right]}^{T}$ is reserved. Otherwise, the next group of candidate sequences is searched. In the search process of this layer, if k candidate sequence has been saved, it is necessary to compare the size relationship between the PED of the new sequence and the maximum PED in the previously saved candidate sequence. If it is greater than the PED of the previously saved sequence, the search continues to the subsequent sequence. Otherwise, the new smaller PED value is saved as a new sequence, and subsequent candidate values continue to be judged to ensure that only k candidate sequences are recorded as long as the largest PED is eliminated. For the candidate sequence determined by the upper layer, the search of this layer can be completed after the search is completed. Then, it can be substituted into the search process of the next layer until the lattice vector the shortest distance from the target vector is obtained after searching all layers, and the shortest vector, $T_{i}{}_{j}(\overset{\wedge }{a})$ can be directly selected as the final solution of the problem. The fixed solution x closest to the floating-point solution y is finally searched, and the schematic diagram is shown in Figure 4.

That is to say, follow the yellow arrow to search from top to bottom, and then from left to right point by point. The blue arrow indicates the Euclidean distance between each search point and the actual point Y. Because the target search point X is the closest to point Y, the red circle with a cross indicates that the Euclidean distance will only increase until the search reaches this point.

## 4. Partial Ambiguity Resolution Experiments

In order to verify the feasibility of this algorithm, different search algorithms were used to analyze the search time of the optimal ambiguity solution based on the MATLAB simulation platform. Construct the vector according to the collected ambiguity variable:(33)$$\overset{\wedge }{a}=100\times randn(n,1)$$

Among them, function randn(n,1) is the function of the MATLAB platform that can randomly generate vectors conforming to a positive distribution.

The simulation case covariance matrices $Q_{aa}^{-1}=UDU^{T}$ and U used in this section are the orthogonal matrices $D=diag(d_{i})$ and $d_{i}=rand$. Figure 5 shows the simulation diagram of the ambiguity search time with dimension 40 by using three algorithms: New-Breadth-First, conventional Breadth-First and LAMBDA. In each process, the time consumed to search the optimal ambiguity value can be calculated.

As shown in Figure 5, the search space was reduced through New-Breadth-First, which significantly shortened the time spent searching for the optimal value compared with the conventional Breadth-First and LAMBDA algorithms. When the dimension was 10–20, the three algorithms had little difference, but when the dimension was large, the New-Breadth-First algorithm could ensure a shorter search time and was better than the other two algorithms. Therefore, according to the numerical simulation results, it can be concluded that the New–Breadth–First algorithm has good effectiveness.

The basic experiment conditions are shown in the Table 1. The data acquisition location is shown in the Figure 6.

Because the carrier phase positioning uses the number of carriers, it is important to obtain it. The fixed solution can be obtained by searching from the floating solution. Figure 7, Figure 8, Figure 9 and Figure 10 show the simulation experiment diagrams of the fixed ambiguity of the conventional PAR algorithm and the New-Breadth-First search algorithm based on the LAMBDA algorithm according to the collected GNSS experimental data. It can be seen from the simulation diagram that the accuracy in all directions was improved by using the New-Breadth-First search algorithm. Compared with the conventional PAR algorithm, the New-Breadth-First search algorithm improved the ambiguity fixation rate by 31.56% and 32.18%. It can be seen that the New-Breadth-First search algorithm has the advantage of fixed ambiguity. The feasibility of the new algorithm is further proven.

It is shown in Figure 8 and Figure 10 that the success ambiguity fixation rates in all directions have been effectively improved by the New-Breadth-First search algorithm, and the positioning accuracy can approach the centimeter level, which further illustrates the effectiveness of the New-Breadth-First search algorithm in high-dimension ambiguity situations.

In order to further verify the fixation effect of the New-Breadth-First search algorithm on the ambiguity in the actual scene, the GNSS observation data (BDS satellites and GPS satellites) collected from the observation station built on the roof of a school library in Guangxi Zhuang Autonomous Region were also used. The ambiguity search algorithm adopted the New-Breadth-First algorithm proposed in this section, which was compared with the conventional PAR algorithm.

## 5. Conclusions

First, this paper introduces the equivalent relationship between lattice theory search and ambiguity search and proposes a new ambiguity search algorithm based on lattice theory. The algorithm is based on the Breadth-First lattice. In the search process, it reduces the ambiguity search space by calculating and judging the Euclidean distance between each search variable and the target one. Thus, compared with the LAMBDA algorithm and the conventional Breadth-First algorithm, it effectively shortens the search time for the optimal ambiguity solution in a high-dimension situation. Combined with the partial ambiguity fixing strategy, the satellites with low elevation angles are eliminated to obtain the optimal subset. The experimental results show that the positioning accuracy of this algorithm is significantly improved compared with the conventional PAR algorithm based on the conventional LAMBDA solution, and the ambiguity fixation rate is effectively improved.

## Figures and Tables

**Figure 1 sensors-22-07126-f001:**
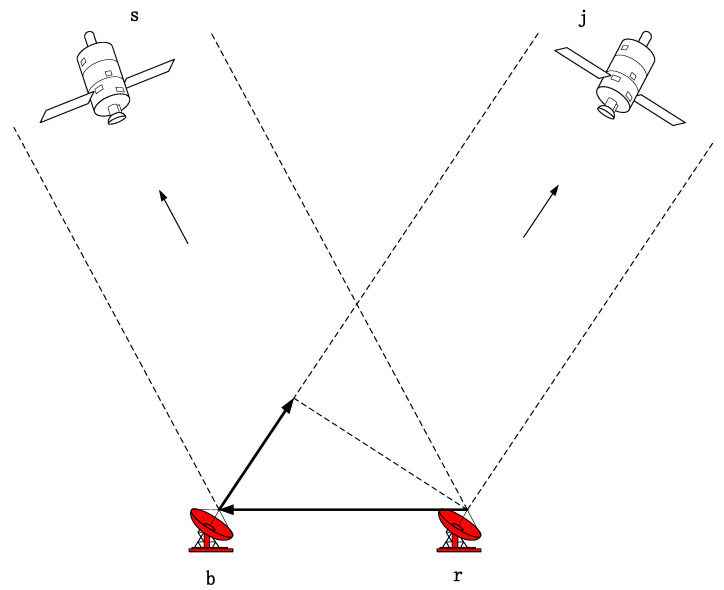
Geometric diagram of double-difference model.

**Figure 2 sensors-22-07126-f002:**
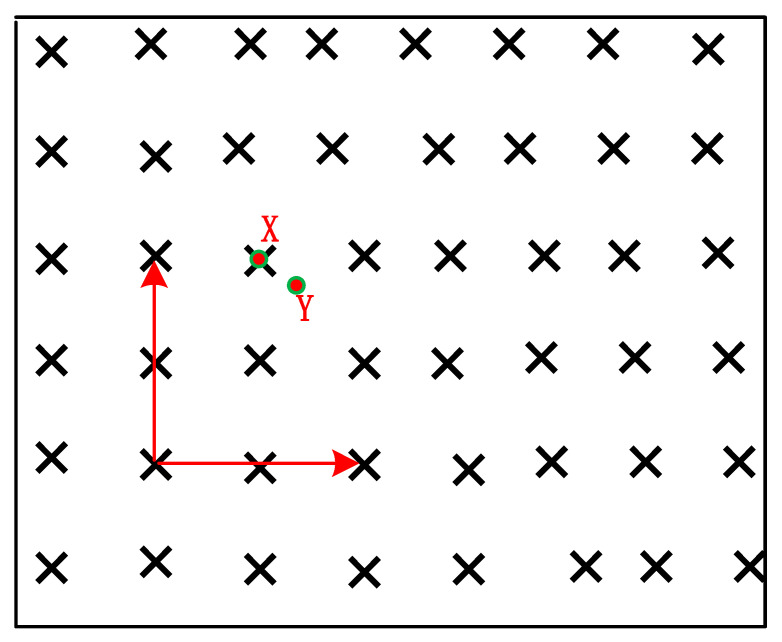
Diagram of nearest grid search.

**Figure 3 sensors-22-07126-f003:**
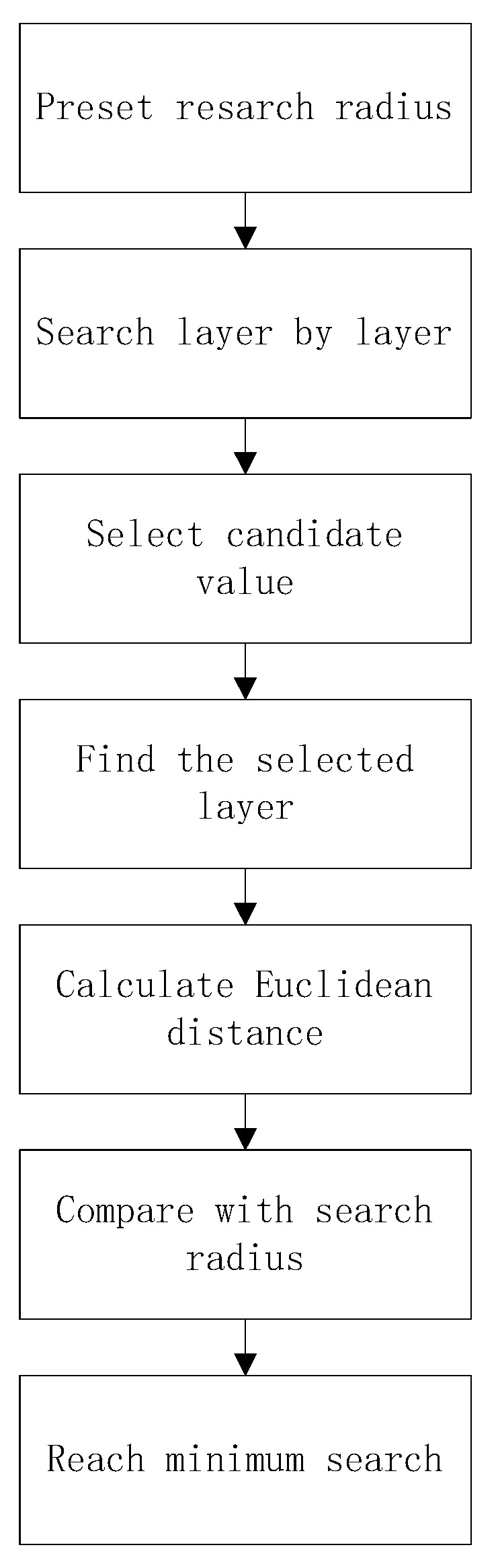
Breadth-first search algorithm flow chart.

**Figure 4 sensors-22-07126-f004:**
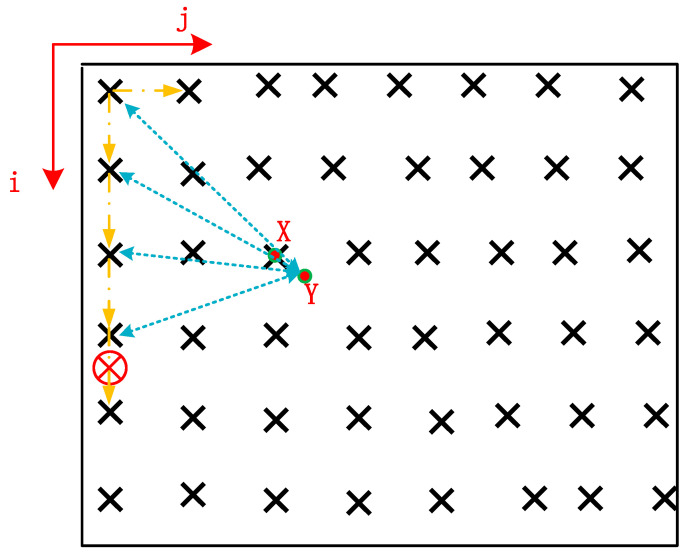
Schematic diagram of ambiguity search optimization algorithm.

**Figure 5 sensors-22-07126-f005:**
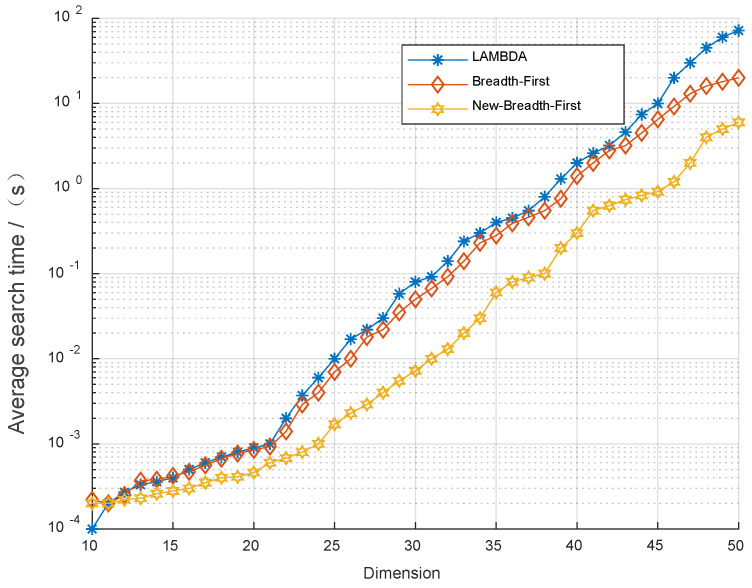
Simulation diagram of ambiguity search time of three algorithms.

**Figure 6 sensors-22-07126-f006:**
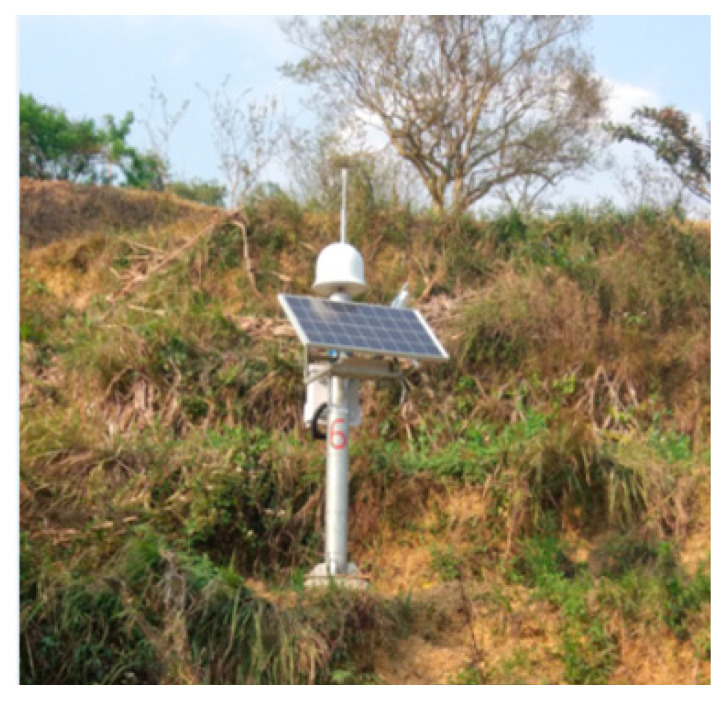
Experimental data collection stations.

**Figure 7 sensors-22-07126-f007:**
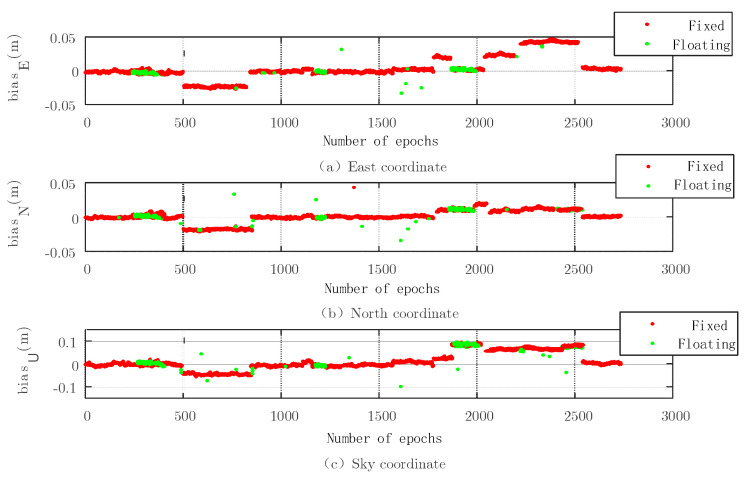
Three−dimensional baseline solution results of conventional PAR algorithm based on the LAMBDA algorithm.

**Figure 8 sensors-22-07126-f008:**
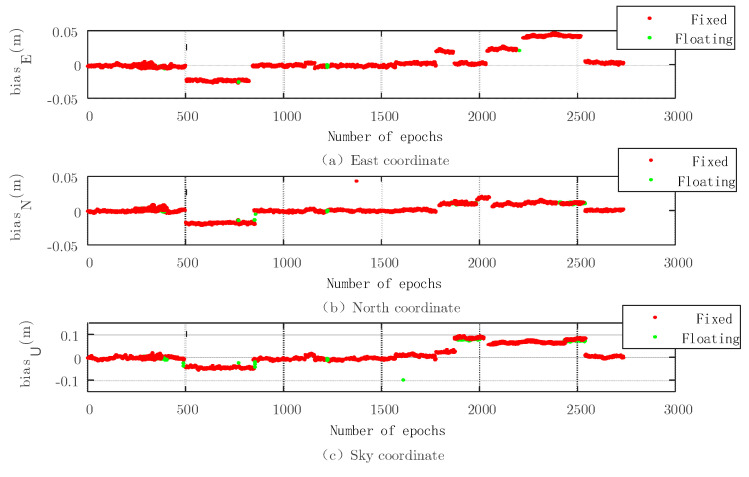
Three−dimensional baseline solution results based on the New-Breadth-First search algorithm.

**Figure 9 sensors-22-07126-f009:**
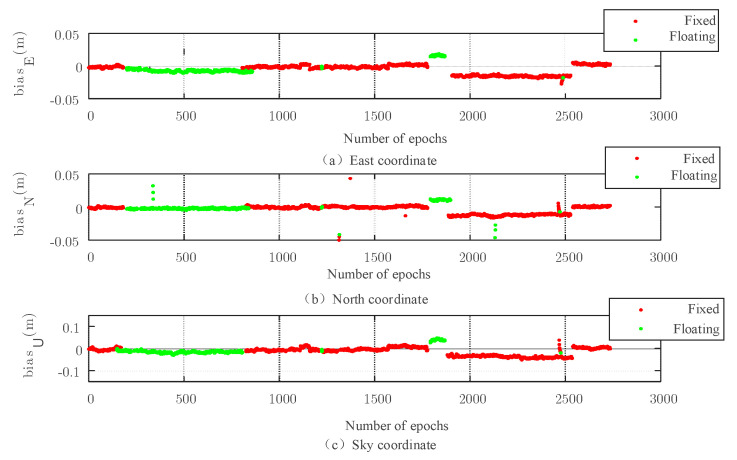
Three−dimensional baseline solution results of the conventional PAR algorithm based on the LAMBDA algorithm.

**Figure 10 sensors-22-07126-f010:**
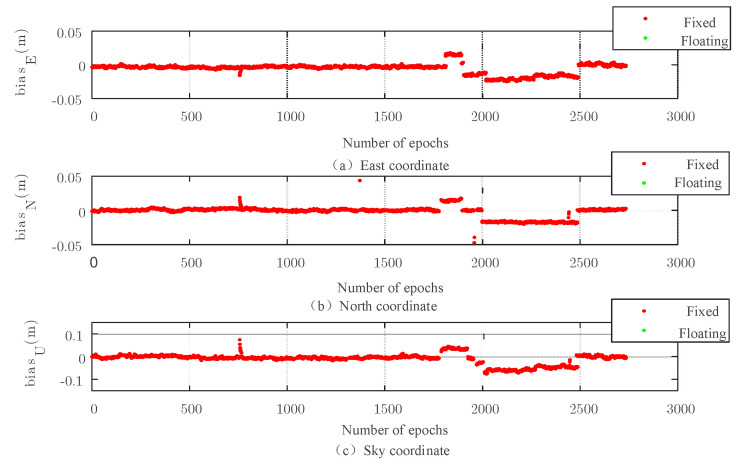
Three−dimensional baseline solution results based on the New-Breadth-First search algorithm.

**Table 1 sensors-22-07126-t001:** Basic information of the experiment.

Receiver Model	ublox-m8
Satellite dimension	30
Data acquisition time	27 October 2020 9:00–11:30
Data format	RTCM3
Epoch number	3000
Experiment solution	Static solution

## Data Availability

Not applicable.

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
