# Peer review of "A Partial Ambiguity Resolution Algorithm Based on New-Breadth-First Lattice Search in High-Dimension Situations"

_sensors, 2022, doi:10.3390/s22197126_

Round 1

Reviewer 1 Report (Previous Reviewer 2)

The authors have made significant corrections and improvements to the paper. However, the method has so far been tested on only a short set of observations. It is commendable that they have repeated the observations and included observations of the same BeiDou.

The paper is suitable for publication in its present form.

Author Response

I have modified and marked in the original text.Thanks.

Reviewer 2 Report (New Reviewer)

The paper applies the breadth-first lattice search algorithm to improve the accuracy of the positioning using Global Navigation Satellite Systems (GNSS). I think the work is technically sound and the mathematical operations are described with enough detail. However, the paper should be improved. I have the following comments:

In equation 21, in the last formula, in the expression under min, where the set of values in the minimization process are defined: maybe α instead c?

In page 9, in lines 209-211 the authors explain that the lattice theory has been applied to many fields. Please provide at least one general reference (like a survey) or several references.

In page 10 the procedure of the breadth-first search mode is explained. It would be very helpful for the reader to add a flowchart of the procedure explained in the text.

In page 11, in line 268 is explained that t is less than or equal to K; I think t is not explained before in the text. In that case, please provide a clear definition of the variable t.

In page 11, figure 3 does not help to understand the breadth-first algorithm. What does the red circle with a cross mean? What do the blue arrows mean? And the yellow arrows? Please explain clearly the procedure shown in Figure 3.

In equation 33, why the standard deviation is set to 100?

In page 13, line 314. Please explain in more detail how the experimental data was collected.

In figures 6, 7, 8 and 9 the results of the traditional LAMBDA algorithm and the New-Breadth-First search algorithm are shown.  What is the difference between figures 6 and 8? The captions are the same; same question for figures 7 and 9. Moreover, please, explain what fixed and floating is.

Additional comments:

The paper must be proofread. There are several sentences which are difficult to understand or have grammar (syntactic) errors. For example: page 1, line 17: “…, and the ambiguity parameters have the characteristics of discretization, …”; page 2, line 42 “…the ambiguity parameters under the high dimension, …”, etc.

Please improve how the variables are shown in the text. For example in page 3, line 67: ‘…established for a receiver r and a satellite i :”, the variable i seems to be written as a superscript; in page 3, line 87, “…between satellite s and satellite j.”, again the variable j is written as a superscript; in page 5, line ”…B is the matrix of non-ambiguity parameters, a is the integer…”, the variable a is either written as a subscript or the font size is wrong, etc.

Please improve also the equations: in equation 17 the variable Qaa^(-1) has two squares on the aa subscript (also in the text).

Author Response

I have modified and marked in the original text. Thanks.

This manuscript is a resubmission of an earlier submission. The following is a list of the peer review reports and author responses from that submission.

Round 1

Reviewer 1 Report

Dear author,

you analysed a method for solving integer ambiguities. This method shortens the search time in relation to the LAMBDA method and to the conventional Breadth-First method.

My major concerns regarding your work are related to the test of the enhencement of the fixing rate. I think your test data set is much to short and maybe more stations would be mor convincing. Probably the fixing rate is already enhenced with the conventional breadth first algorithm? And the advantage of your methode is only calculation time. So it would be much more interesting to show the time benefit of your algorithm on real data. Using different number of satellites GPS only, GPS+Beidou, GPS+Beidou+Galileo, .... If I am wright please also adopt the conclusions in this way.

Addiononally:

Abstract: line 14 an and is to much

Introduction: You repeat the abstract in the first paragraph, this is not very interesting for the reader. Please also rephrase  line 28 to 34 of this paragraph and put it into better English.

Chapter 4 line 112: you mention a famous question, but you do not formulate it as a question

Formular 15: I think the second y-Ba is a y-Ra, or?

The B in this equation is not the same as in equation 4, or?

line 205 you mention 3 steps but there are 4

In the hole text please use capital letters for the names of the methods.

Author Response

Hi:

I have modified the text and marked it in red font. Please check it.

Author Response

(The authors gave the same response as above.)

Reviewer 3 Report

This paper proposes a new ambiguity search algorithm based on lattice theory, that can shorten the search time compared with conventional method. The paper is well-organized, but there are several points needs to be improved:

1. The background of lattice theory is needed, otherwise section 5 and section 6 is hard to understand for readers who are not experts in this area.

2. In line 208, the section index is wrong, it should be section 6.

3. The notation in this paper needs to be adjusted. For example, "B" is used for both lattice base and coefficient matrix of non-ambiguity matrix parameters in equation 4. This would cause confusion. There are also other notations which have been multi-defined. Please check and make edits.

4. In line 41, a "in" is missing.

5. In line 44, "In this paper, Chang [2] reduced..." Is the contribution comes from this paper or the cited one?

6.In line 46, "insert" should be "inserted".

7.In equation (15), min sign is missing.

8.In equation (31), "=" is missing.

Author Response

(The authors gave the same response as above.)

Round 2

Reviewer 1 Report

Dear author,

there is no reply to my major comments. I do not think that this study can be published if the analysis is not extended further.
